# Changing language input following market integration in a Yucatec Mayan community

Cecilia Padilla-Iglesias[ID][1,2]*, Amanda L. Woodward[3], Susan Goldin-Meadow[3], Laura A. Shneidman[4]*

1 Department of Archaeology, Leverhulme Centre for Human Evolutionary Studies, University of Cambridge, Cambridge, United Kingdom, 2 Institute of Anthropology, University of Zurich, Zurich, Switzerland, 3 Department of Psychology, University of Chicago, Chicago, IL, United States of America, 4 Department of Psychology, Pacific Lutheran University, Tacoma, WA, United States of America

* cecilia.padillaiglesias@uzh.ch (CPI); las@plu.edu (LAS)

**Data Availability Statement:** All relevant data are within the manuscript and its Supporting Information files.

## Abstract

Like many indigenous populations worldwide, Yucatec Maya communities are rapidly undergoing change as they become more connected with urban centers and access to formal education, wage labour, and market goods became more accessible to their inhabitants. However, little is known about how these changes affect children's language input. Here, we provide the first systematic assessment of the quantity, type, source, and language of the input received by 29 Yucatec Maya infants born six years apart in communities where increased contact with urban centres has resulted in a greater exposure to the dominant surrounding language, Spanish. Results show that infants from the second cohort received less directed input than infants in the first and, when directly addressed, most of their input was in Spanish. To investigate the mechanisms driving the observed patterns, we interviewed 126 adults from the communities. Against common assumptions, we showed that reductions in Mayan input did not simply result from speakers devaluing the Maya language. Instead, changes in input could be attributed to changes in childcare practices, as well as caregiver ethnotheories regarding the relative acquisition difficulty of each of the languages. Our study highlights the need for understanding the drivers of individual behaviour in the face of socio-demographic and economic changes as it is key for determining the fate of linguistic diversity.

## Introduction

Like many indigenous communities throughout the world, Yucatec Maya communities in Mexico are undergoing rapid change as they integrate into dominate market economies [1–7]. In this paper, we consider how these changes may affect children's experiences, by asking whether the language addressed to children shifts during a period of ongoing market integration and why it might do so.

There are several reasons to predict language input change. First, during market integration (and associated opportunities for wage labor, access to education and market goods), parents may shift input away from minority local languages and towards majority surrounding

**Funding:** This study was funded by the Department of Archaeology at the University of Cambridge and by a NSF award (BCS-1226113) to Amanda Woodward.

**Competing interests:** The authors have declared that no competing interests exist.

languages because they believe that dominant languages are socially and economically advantageous for children to learn [3–10]. For example, Hill and Hill [11] and Rolstad [12] attribute the decline of Nahuatl speakers in Mexico over the 20th century to parents using Spanish (rather than Nahuatl) with their children in order to prepare them for the newly introduced bilingual schooling policies. But changes in the input received by children in small-scale communities could also be caused by factors beyond those that imply parental devaluation of native languages.

Market integration could lead to changes in childcare practices that shift caregiving away from older children to adults and consequently affect language input [13,14]. In many cultural settings, it is *not* parents who have traditionally served as infants' primary interlocutors, but rather other children [15–17]. If changing market economies result in infants spending less time with older siblings, this could result in less time that infants are engaged in directed conversations overall (including conversations in a local language). Changes in childcare in emerging integrated contexts could result from several factors. For example, the reduction in fertility that often accompanies transitions from subsistence to skill-based economies could mean that there will be fewer older children to care for, and talk to, young children [14,18]. Moreover, if older children are expected to devote more time to their education, they may have less time to devote to household labour, including childcare [19,20]. Consequently, speech directed to young children might face overall decline, and, when combined with the other factors described below, affect local language input, even if there is no change in parental beliefs about the value of one language over another. Given the pivotal role of child-directed speech for the subsequent development of linguistic competences [21], these changes in input patterns could gradually result in fewer and fewer members of the younger generations becoming proficient in the local language.

Additionally, during the process of market integration, both parents and older siblings of young children in small-scale communities are more likely than previous generations to have had access to formal schooling provided by the majority community. This access both exposes caregivers to non-local languages (providing them with the ability to use the language in interaction with young children) and also introduces caregivers to the interaction patterns of the dominant culture. Schools generally model a pattern of interaction and socialization in which children are directly addressed in pedagogical interaction (e.g. [22]) Caregivers who have been schooled might integrate these socialization patterns into their childcaring practices with respect to both their local and non-local languages [14,23–25,28], by, for example, directing more speech to children. Alternatively, they might adopt pedagogical practices more narrowly, reserving the non-local language (also the language of school instruction) for directed interactions (those typical of school contexts). If so, directed speech to children would occur more often in the non-local language than the local language, again potentially affecting the relative ability of children to learn each of the languages.

Finally, broader cultural ethnotheories about language learning may affect caregiver's input strategies during market integration. Rather than undervaluing the local language, parents may use directed speech specifically in a non-local language because they have beliefs about the necessity of doing so. Previous research suggests that, in many small-scale communities at greatest risk for language loss, development of the native language is thought to come from within the child, requiring little outside intervention [1,8,16]. In these contexts, directed linguistic input may be viewed as less important for supporting native language transmission than it is in cultural communities where directed teaching is considered critical [26–28]. Indeed, Shneidman and Goldin-Meadow [16] found that caregivers from the United States were 7 times more likely to direct speech to their infants than caregivers from rural Yucatec Mayan communities. Other studies have confirmed this general pattern––infants from

market-economies are between 2 and 11 times more likely to receive child-directed speech (CDS) than infants from small-scale, subsistence economies [17,29,30].

Along the same lines, caregivers may believe that the non-native language is more difficult to learn than the native language, leading to a need for more focused input in the non-native language. In emerging bilingual contexts adults and older children are more likely to have learned the non-local language later in life and thus may have encountered difficulty in their own learning. This may lead them to the false conclusion that the non-local language is inherently more difficult to learn than the local language [3]. Such a belief could result in caregivers using more directed speech to children in the non-local language than the local language, even when they desire that children acquire equal competences in each.

In summary, four factors could explain the fact that children receive less directed speech in their native language than in the non-local language when small-scale societies are being integrated into majority cultures: 1) a general devaluing of the native language, 2) changes in child-care practices, 3) changing access to education and 4) specific parental beliefs regarding the importance of directed input for language learning. In order to assess the contributions of these factors, it is necessary to quantify changes in children's input during periods of market integration, and determine how these changes intersect with changing demographics and belief systems. This is not only important to broaden our understanding of the consequences of market integration for indigenous communities but to predict the fate of the world's linguistic diversity. As proposed by Lambert [10], the sustainability of minority languages might be unavoidably compromised where bilingualism is a side-effect of a general process by which individuals feel forced or incentivized to put aside their indigenous language for a more necessary or prestigious one. However, there is also the possibility of "additive bilingualisms" whereby individuals may want to invest in learning an additional language to be able to communicate with out-groups for particular purposes (such as education, commerce or travels) whilst retaining their ethnic (and linguistic) affiliation.

We address these questions by taking advantage of a "natural experiment" arising in the Yucatán peninsula as a result of economic development, new means of transportation, opportunities for wage labor, and access to education. Using video data collected from natural interactions, we compared the linguistic input received by 16-24-month-old infants growing up in the same Yucatec Mayan villages across two cohorts approximately 6 years apart (n = 21 in 2007/08, see ref. 20, and n = 15 in 2013/14). Previous work in this community showed that input heard by infants in 2007/2008 was primarily in the local language, Yucatec Maya, and provided by other children rather than adults [16]. Here, we ask whether these characteristics shifted in this community after a period of increased market integration. In addition, we conducted structured interviews with 126 adults from the villages (including all of the infant's primary caregivers) in January 2019. These interviews provide demographic information related to the changing economy, educational practices, and care practices, and also shed light on caregiver beliefs and attitudes about language. In particular, they provide insight into how important caregivers thought Maya vs. Spanish was for their infant's future success, and how difficult they thought the two languages were to learn. We examined whether language input changed over the period of market integration––in quantity, type (directed, overheard), and language choice (Maya, Spanish)––and whether those changes in language input were related to childcare, educational, and belief practices.

## Results

We first used interview data and family histories to assess socio-demographic changes across the six-year period in the focal villages. As expected, there were general differences in variables

related to increased contact with the majority culture across cohorts––greater access to education in the majority language, wage labor, and market goods (S1 Text, S1 and S2 Tables, S1 Fig). For example, in 2007, children could only complete primary school in their natal village; by 2013, they could also attend secondary school. Primary school attendance of caregivers had also doubled across cohorts (2007: Mean = 2.17, SD = 1.64; 2014: Mean = 4.09, SD = 2.51). Likewise, whilst 66% of fathers from infants in cohort 1 engaged in some form of wage labor (regardless of whether they also worked in agriculture), 82% of those from infants in cohort 2 did so. In cohort 1, the vast majority of fathers (89%) had a "milpa" (i.e. worked in agriculture) regardless of whether they also engaged in wage labour. However, this number had dropped to 63% in the second cohort, indicating that more households are transitioning to a complete reliance on market jobs. However, in line with findings by Schacht et al. [31] and Gaskins [1], many aspects of the villages had not changed during this period: Age of mother at first birth (2007: Mean = 20.08, SD = 3.03; 2014: Mean = 19.60, SD = 4.22), and average number of older siblings of focus infants (2007: Mean = 3.89, SD = 2.75; 2014: Mean = 3.19, SD = 2.52); no infant had younger siblings because the target infant was always the youngest of the family (weaning age ranged from 2–2.5 years).

In order to evaluate whether there had been changes in patterns of linguistic socialization, and the prevalence of Spanish in the speech heard by infants following market integration, we used transcriptions from the natural recordings. First, we assessed the number and type (child-directed, overheard) of utterances received by infants, and who produced the utterance (primary caregiver, another adult, older child). We fitted Bayesian multilevel zero-inflated Poisson models to the data, and found changes across cohorts in all three measures (see S1 Text). Mean number of utterances per hour directed to children decreased from 2007 to 2014 from primary caregivers (99.6 to 76.8 utterances) and, particularly, from other children (346.7 to 165.7), The number of directed utterances from other adults remained relatively stable (36.86 to 37.79) (Fig 1, top three graphs). In contrast, mean number of overheard utterances per hour increased slightly in primary caregivers (29.8 to 51.0) and in other adults (38.9 to 66.5), and decreased slightly in other children (104.6 to 93.9; Fig 1, bottom three graphs). For all input types, most utterances originated from other children (68.8% of utterances in cohort 1, 52.8% in cohort 2; S7 Table). The results were consistent across all villages (i.e. village-specific intercepts were roughly symmetrical around 0; S2 Fig). Thus, the overall amount of directed speech children heard, particularly from other children, decreased over the 6-year period; in contrast, the amount of overheard speech children heard changed very little.

While there was no significant difference in age between cohort 1 and cohort 2, the age ranges of both cohorts were not completely overlapping (the age range for cohort 1 was 16.1–24 months and cohort 2 was 16.1–22.5 months). Because older children might elicit more speech from their caregivers, due to changing language competences, we assessed whether the decrease in overall input (and in particular input from primary caregivers) from cohort 1 to cohort 2 could be explained by child production or child age. We first fitted the same Bayesian multilevel zero-inflated Poisson model with "Number of directed utterances per hour received by target child from primary caregiver" as response variable and "Cohort" as well as "Number of utterances produced by the focal child" as predictor variables. Consistent with our previous findings there was a negative effect of "Cohort" on the number of utterances that infants received from their caregivers (Estimate = -0.43; 90%HPDI: [-0.53,-0.33]). However, the number of utterances produced by the focal child did not affect the number of directed utterances they received from their caregivers (Estimate = -0.01; 90%HPDI: [-0.01,0]). We next re-fitted all the models reported in Fig 1 and excluding from Cohort 1 all 23- and 24-month-old infants (N = 2). Results (see S3 Fig) showed that, without these children, the apparent reduction in CDS from the primary caregiver disappeared, perhaps indeed owing to the fact that mothers

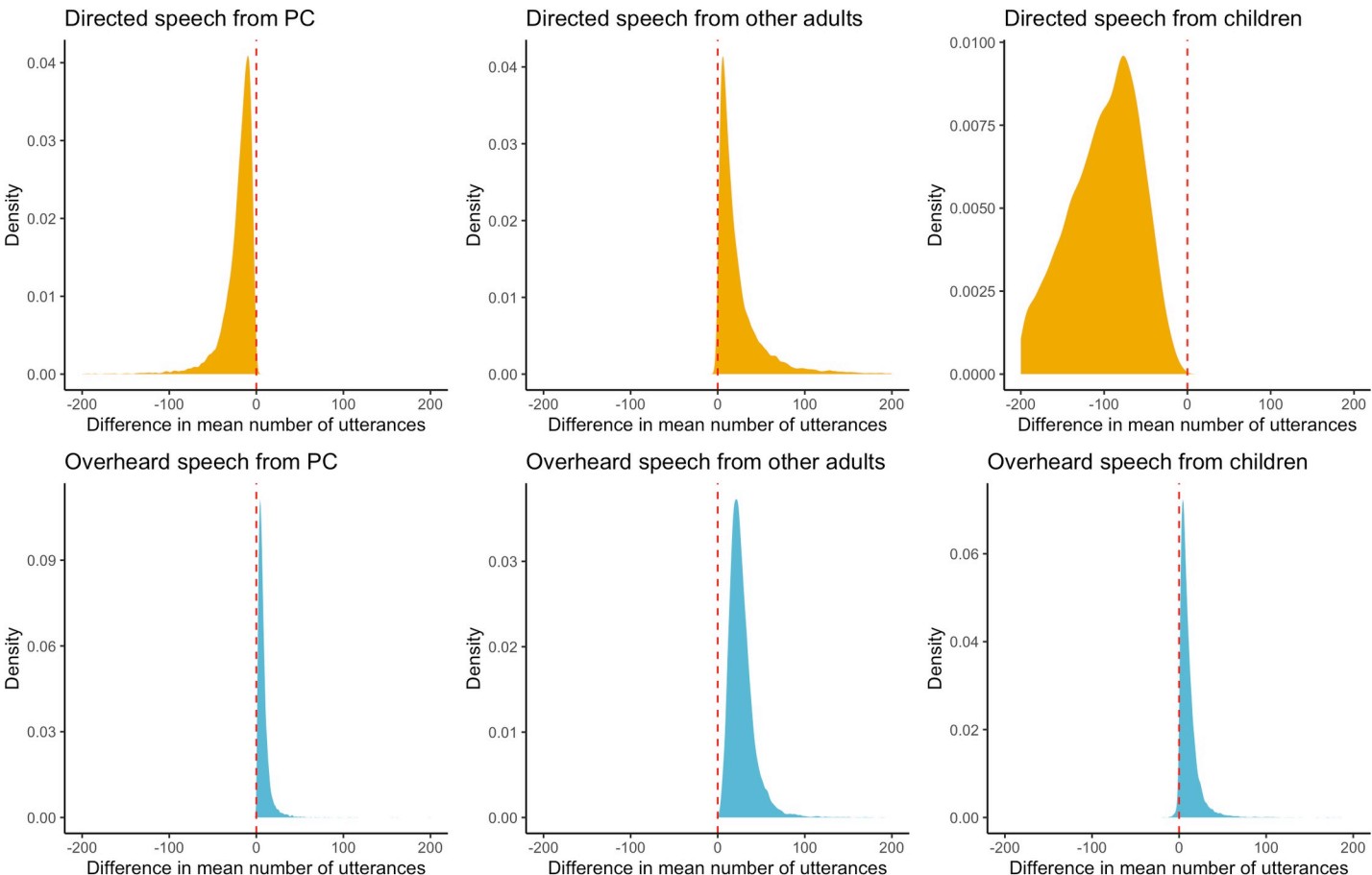

**Fig 1. Posterior predictive distribution of the mean difference in number of utterances of each type received by the average child from 2007 to 2013, as obtained from the Zero-Inflated Poisson model including "Cohort" as predictor variable and the number of utterances of each type as response variable.** These were obtained by averaging from 12000 samples from the posterior distribution (setting the standard deviations for the varying intercepts to 0). From top-left to bottom-right: Directed input from primary caregiver, directed input from adults, directed input from children, overheard input from primary caregiver, overheard input from adults, overheard input from children.

are more likely to direct speech to older infants. However, there was still a significant overall reduction in the total number of directed utterances received by infants from the second cohort, stemming from a reduction in utterances directed from other children. Thus, even when taking child production and age into account, infants from cohort 2 received less directed input than infants from cohort 1.

To examine whether the language of input shifted from Yucatec Maya to Spanish over the 6-year period, we fitted Bayesian logistic mixed models to the proportion of input in Spanish (vs. Maya) that infants received in child directed and overheard speech. Since code-switching within utterances was extremely rare (mean = 1.6 utterances per hour, SD = 2.08 and mean = 3.5 utterances per hour, SD = 7.09 in cohorts 1 and 2 respectively; out of an average of 464 utterances heard per hour) and did not vary across cohorts (t = -1.01, df = 14.59, P = 0.33), we excluded them from these analyses. We found significant increases in the proportion of child-directed input that infants received in Spanish (instead of Yucatec Maya) between 2007 (Mean = 0.21, 90% HPDI: [0.03, 0.47]) and 2014 (Mean = 0.67, 90% HPDI: [0.32, 0.93]) (Fig 2, left graph). In contrast, there were no significant changes in the proportion of overheard speech infants received in Spanish between 2007 (Mean = 0.27, 90% HPDI: [0.07,0.55]) and

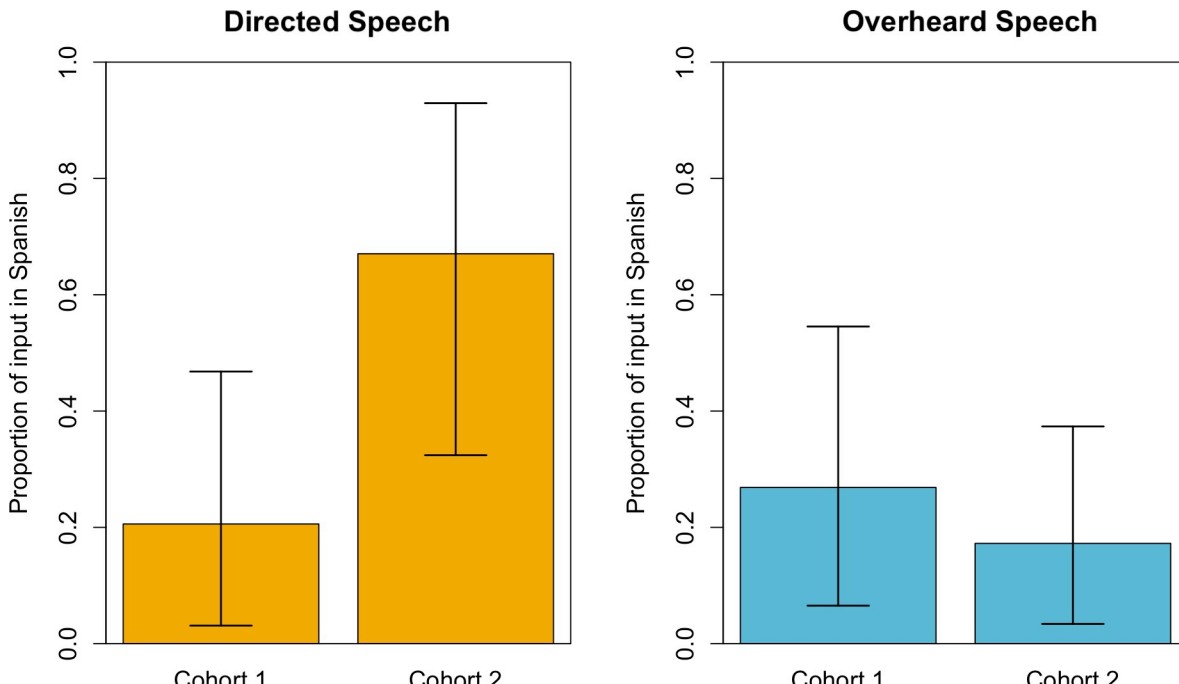

**Fig 2. Posterior means (bars) and 90% HPDI (error bars) for the models predicting the proportion of utterances in Spanish (Number of utterances in Spanish out of the total number of utterances of that type) an average infant (infant with an intercept at 0 for village) is exposed to.**

2014 (Mean = 0.17, 90% HDI: [0.03, 0.37]) (Fig 2, right graph). Thus, caregivers altered the proportion of speech they directed to infants in favor of Spanish over the 6-year period, but did not change the proportion of speech overheard by infants (S8 Table). By 2014, caregivers (whether they were primary caregivers, other adults, or children; S8 Table) used Spanish in the speech they actively directed to infants, but primarily used Maya in the speech overheard by infants.

To summarize, infants in cohort 2 generally received less directed input than infants in cohort 1 but, when that input was addressed to them, it typically came in Spanish. Indeed, changes in directed input were driven, fully, by decreases in directed input in Yucatec Maya (S12 Table). When we consider the total number of utterances per hour in directed speech heard by infants in each of the languages, we find that infants in cohort 1 heard an average of 397.3 Maya utterances per hour in directed speech, but infants in cohort 2 children heard only an average of 92.51 Mayan utterances. Why did caregivers decrease the amount of Maya they directed to infants? We turn to interview data gathered in 2019 to address this question. 126 adults from these villages (including all of the primary caregivers from the two cohort samples) were asked if it was more important to learn Maya or Spanish and, as a follow up question, why each language was important to learn (S1 Text). All of them reported that learning Maya and learning Spanish were equally important. Responses to questions about why it is important to learn Maya and Spanish are summarized in Table 1. In general, Spanish was seen as a functional tool important for necessities, such as communicating with doctors or going on shopping trips to nearby Spanish speaking cities. Maya, in contrast, was regarded as important for maintaining social relationships and cultural ties within the village.

In order to explore attitudes about the need for directed input in learning language, adults were asked: "Some people believe that language needs to be taught, while others believe that

**Table 1. Answers to questions regarding why each language is important from interviewees.**

| Question | Answer | Count |
|---|---|---|
| **Why is Maya important/necessary?** | To communicate in the village, as everyone speaks Maya | 34 |
| | To preserve the language/prevent disappearance | 18 |
| | Some people don't understand Spanish | 13 |
| | To know where one comes from, it is our cultural heritage/tradition | 9 |
| | One needs it for everything | 4 |
| | Learning Maya allows you to learn how to speak earlier in life | 3 |
| | It is pretty | 3 |
| | For work (requisite in hotels) | 2 |
| | Help old people translate | 2 |
| | To speak to one's family | 1 |
| | It is more similar to English | 1 |
| | To teach one's children | 1 |
| | If you don't, people here make fun of you | 1 |
| | To go shopping | 1 |
| | To be able to work in corn fields ("milpa") | 1 |
| | **TOTAL** | **96** |
| **Why is Spanish important/ necessary?** | For going to the doctor | 31 |
| | For visiting cities | 30 |
| | To speak/understand visitors | 16 |
| | To fully express yourself | 7 |
| | To be able to work outside | 13 |
| | Now one needs to learn it for everything | 6 |
| | One needs it to talk to schoolteachers | 6 |
| | It is very popular now | 6 |
| | For understanding telenovelas | 1 |
| | For bureaucracy | 1 |
| | In case you get lost | 1 |
| | **TOTAL** | **118** |

Note that counts do not add up to 126 because responses that did not answer the question were excluded as they offered no explanatory power.

language will just come out on its own; what do you believe? And, as a follow-up, "Is that the same for Maya and Spanish learning?" In response to the first question, 81.9% (N = 98) of adults believed that infants needed to be actively instructed in how to speak. This pattern was found in the primary caregivers of infants in both the first (72.3%; N = 16) and second (60.9%; N = 9) cohorts, with no significant differences between cohorts ($\chi^2$ = 0.20, df = 1, $p$ = 0.65). As an example, one mother stated: "Mothers need to speak to their babies for them to learn how to speak because alone they can't. I have a nephew who only learned how to speak when he was seven because his mother didn't teach him." Interestingly, in response to the second question, 49.4% (N = 61) of the interviewed adults believed that infants learnt Spanish and Yucatec Maya in different ways (S11 Table). Specifically, 34.1% (N = 43) of interviewees thought Yucatec Maya was easier to learn than Spanish, sometimes reporting that Spanish required infants to be actively taught, whereas Maya could be learned from overhearing. This pattern raises the possibility that an additional factor responsible for the increase in directed speech in Spanish arose, not from devaluing Maya (and thus trying to substitute Spanish for it), but from the belief that child-directed speech is essential in order to learn Spanish, but not Maya.

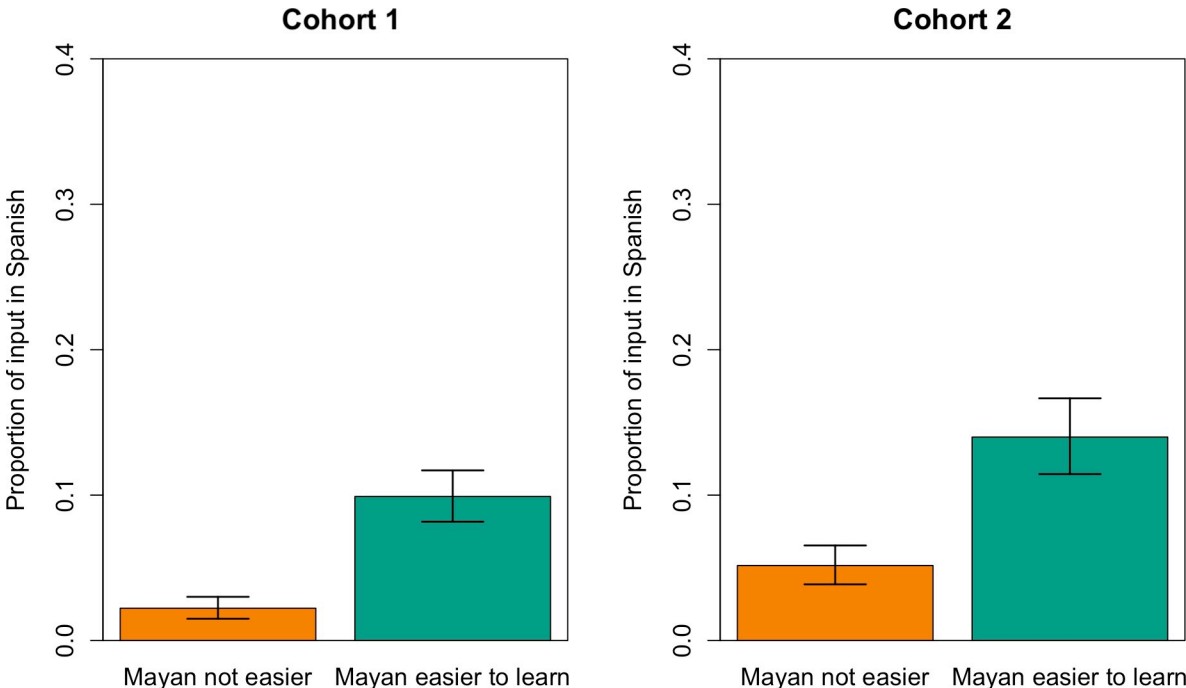

**Fig 3. Posterior means (bars) and 90% HPDI (error bars) for the models predicting the proportion of utterances in Spanish (Number of directed utterances in Spanish out of the total number of directed utterances) primary caregivers directed to their infants in Cohort 1 (left) and Cohort 2 (right) according to whether they believed Yucatec Maya was learnt easier/faster than Spanish or not.**

To test this hypothesis, we coded caregivers' responses to the above question based on whether the answer suggested that the respondent thought that Maya was easier to learn than Spanish, Spanish was easier to learn than Maya, the two languages were learnt in the same way, or the response was not codable along this dimension (e.g., "They learn Spanish better when they are older") (See S11 Table for details on the coding of answers). We then used Bayesian logistic mixed models to assess whether the belief that Spanish was harder to learn than Maya predicted the proportion of input that caregivers directed to their infants in Spanish. We confirmed that children whose primary caregivers thought Spanish was harder to learn were more likely to speak to their children in Spanish both in 2007 (log-odds = 1.59, 90% HPDI: [1.06, 2.1]) and 2014 (log-odds = 1.1, 90% HPDI: [0.67, 1.56]) (Fig 3). The proportion of mothers who held the belief that Maya was easier to learn was not different across cohorts (58.3%, n = 7 versus 33.3%, n = 3; Fisher's exact text, $p = 0.21$), nor was it related to how fluent they were in Spanish ($\chi^2 = 0.47$, df = 1, $p = 0.50$).

## Discussion

Our results show, for the first time, that the quantity, type, source, and language input to Yucatec Maya infants changed as the communities became more connected with urban centers, and hence education, wage labour, and market goods became more accessible to their inhabitants. Infants in cohort 2 received less total directed input than infants in cohort 1 and, when they were directly addressed, most of their input was in Spanish. The net result of these changes was that infants in cohort 2 received far less directed input in the Yucatec Maya language than infants in cohort 1.

Theorists have proposed that shifts in input away from minority local languages and towards majority languages are due to a belief that dominant languages are more advantageous

for children to learn than the minority language [8–10]. However, in the Yucatec Mayan community considered in this study, reductions in Mayan input did not seem to be due to speakers devaluing the Maya language––our interview data suggest that caregivers saw value in infants learning *both* Maya and Spanish, but for different reasons [32]. Spanish was seen as important for achieving pragmatic goals, such as going to town, finding work, or going to the doctor; Maya was seen as important for social goals, such as communicating in the village and maintaining the Mayan culture. One marker of the continued importance of Maya was the fact that, across cohorts, most overheard speech was in Maya throughout the period of change. This pattern indicates that Spanish was not replacing Maya overall, but only in speech directed to children.

We hypothesize that several factors, other than devaluing the local language, may contribute to the decreasing amounts of input in Maya that infants received. First, in this community, both in the past and currently, most input to infants comes from other children, not adults. We found an overall reduction between cohorts in child-directed input from other children, which might have resulted from infants spending less time with other children. These reductions are unlikely to be due to shifts in family composition following increased market integration (as has been suggested in other small-scale communities, e.g. [2,10,20] because we found no evidence that family size changed across cohorts (S1 and S2 Tables). It is possible, however, that schooling has gained importance over this period of time and that older children are focusing more on schoolwork and less on sibling care. Indeed, in a comparison of childcare patterns in 1992 and 2011 in a very similar Mayan community, Kramer and Veile [33] report that, even in the absence of changes in fertility or residence arrangements, the amount of time 7–10 year-old children dedicated to childcare decreased from 12% to a little over 6% (primarily traded off by care from fathers). In future research, we plan to evaluate whether similar changes in childcare practices have occurred in the target communities and, if so, why. Although we also observed a slight yet significant decrease in the quantity of child-directed input infants received from their primary caregivers in the second cohort when compared to the first one, this decrease can be explained by the presence of older children in our cohort 1 sample than in our cohort 2 sample.

The second factor that may contribute to the decreasing amounts of input in Maya that infants received is caregiver beliefs about the importance of directing speech to children in Maya vs. Spanish. We found that many caregivers endorsed a belief that Maya is more 'naturally' or 'easily' acquired than Spanish; parental effort is therefore better placed at teaching Spanish. This result mirrors findings from other indigenous communities in the Americas. For example, following the introduction of bilingual education programs in a K'iche Mayan community, Choi [34] found that many families did not support teaching K'iche at home or in school. The caregivers thought that the K'iche language was "naturally" learned without schooling, and therefore any attempts to teach it would be a waste of their and their children's time. However, the same attitude did not apply to learning Spanish; the caregivers believed education to be necessary for Spanish to be learned. Several other researchers working in bilingual indigenous communities have reported similar beliefs (e.g. [8,35]).

Our results demonstrate that these beliefs matter for predicting caregiver behavior. Caregivers who thought that Maya was easier to learn than Spanish were more likely to use Spanish, and less likely to use Maya, when speaking directly to infants (Fig 3). Previous research indicates that speech directed to infants, and not overheard speech, predicts children´s later language competences [36]. This result holds even in Mayan communities where overheard speech is prevalent [16]. Given this finding, one possibility is that differential ethnotheries regarding language acquisition could cause majority languages to replace minority ones despite the fact that caregivers place equal value in both languages. Nonetheless, whether active

valuation of the native language results in the majority language replacing the minority language, or in the coexistence of both tongues, is an important issue for future research.

Future research should also consider the role of the child's production in influencing the amount of directed input that he or she receives. Although, in the current study, we did not find a relation between the amount of talk that children produced and the input they received from others, we did not conduct a more fine-grained assessment of children's competencies in each of the languages. It could be the case that, if decreases in the input directed to children are language-specific (i.e. in Maya), children may become less competent in Maya, and that this decrement might then affect the amount of Maya the children elicit in directed input.

In conclusion, in the Mayan communities that were the focus of this study, increased market integration over the past decade has been associated with an overall decrease in speech addressed to infants, as well as a decrease in the proportion of total input that infants hear in Yucatec Maya, which is replaced by Spanish. Our results suggest that Maya caregivers are not actively trying to replace Yucatec Maya with Spanish because they believe that Spanish is more social or economically invaluable than Maya. Rather, we hypothesize that caregiver perceptions of the relative difficulty of learning Spanish vs. Maya, as well as changes in childcare practices, are factors associated with decreases in native-language input directed to children.

## Materials and methods

### Ethnographic context

All the data come from four Mayan villages located in the state of Yucatán about 80 miles to the southwest of Cancun (the largest Spanish-speaking urban centre in the region) (S1 Text). Traditionally, Maya families have made their living as subsistence maize farmers. Following the Mexican Revolution, the ejido land tenure system was established, whereby each Mayan village is conceded a plot of land on which to build a house, as well as surrounding country for crops, pasture, and woodland [37]. Ejido lands (as initially written) could not be owned, inherited, sold, or rented and their dominion resided within the village collective, which distributed them among married males [38].

Recently the Maya have experienced rapid socioeconomic changes due to the growth of lowland towns, creation of new roads, improved transport, greater availability of schools, and increased contact with Mexican and global cultures [38]. These changes allowed many individuals, particularly unmarried males, to work for wages in nearby Spanish-speaking urban centers, such as Cancun or Playa del Carmen [39]. However, wage jobs are seen as a supplement to agricultural work, needed to increase household productivity in times of need and not as a replacement for it (S1 Text). Nonetheless, in contrast to what many (e.g., [40,41]) have regarded as an inevitable and gradual process of acculturation, social and residential structures seem to have remained strikingly stable [1].

Another important change in the region has been the increased access of children from rural communities to education. In most schools (including those from the villages of the present study), textbooks are provided only in Spanish and, although some teachers do use both languages, Maya is principally employed for classroom management, whereas Spanish is the language of instruction [42].

### Naturalistic video recordings

Families that contained a target infant between 16–24 months were video recorded for 60 minutes in their homes in natural interaction by the last author who has worked in the villages since 2006 and is familiar to each of the participating families. The first cohort of recordings

was obtained in 2007–2008 [16] and the second cohort in 2013–2014. In total, 36 recordings of 29 infants were analyzed; 21 recordings in the first cohort (mean age = 20.88 months; SD = 2.75; 50% female) and 15 in the second (mean age = 18.17 months; SD = 1.35; 36% female). In both cohorts, an experimenter followed target infants wherever they went (see [43] for details). Family members were instructed to act as they would have had the experimenter not been present and were told that the infant was permitted to go anywhere that they normally would be permitted to go. Recordings varied in their location (inside and outside) depending on the actions of the infant and family members. Infants and family members typically engaged in activities like exploration outside, visits to neighboring households, eating, playing, grooming, etc. It was not uncommon for persons outside the nuclear family to appear in the video recordings (nearby extended family, shopkeepers etc.).

7 infants from the first cohort had more than one video. However, the recordings for those infants with multiple videos were never less than 6 months apart. Since we used hierarchical models (see Statistical Approach), we are confident that including more than one video per infant has not led to biases in inference. One female head of household had infants recorded in both cohorts.

## Transcription and coding of naturalistic recordings

All audible speech from the video recordings was transcribed by local bilingual Yucatec Maya-Spanish speakers (who personally knew the children and their families) and divided into utterances, using the same criteria as Shneidman and Goldin-Meadow [16].

Each utterance was classified by the first and last authors on the basis of:

a. Who was speaking, primary caregiver (mother), other adult (over 11 years), or other children.

b. Whether it was directed to the infant or overheard by the infant. Speech was considered directed if it was addressed to the infant alone or if it was addressed to a group of individuals that included the infant. All other speech was categorized as overheard. Several cues were used together in order to categorize utterances as directed or overheard: Gaze direction, grammatical marking, utterance content, and proximity to infant.

c. The language in which the utterance was delivered. Utterances were classified as Spanish, Maya, or Code-Switch if speakers switched between languages mid-utterance. Because many object words (e.g., *coche*), nicknames (e.g., *Gordo*), kinship terms (e.g., *tía*), numbers greater than three (e.g., *cinco*), and calendar terms (e.g., *Enero*) are borrowed from Spanish even by monolingual Maya speakers, utterances containing these loan words (either embedded in a Maya utterance or alone) were classified as Maya (and not as a Code-switch or Spanish utterance). Only if speakers chose to use a Spanish word when there was a commonly used Maya alternative were those words and utterances classified as Spanish (for example if a speaker said "nariz" for "nose" instead of the Maya word "ni").

The classification of all utterances according to the criteria a–c above was done from the transcriptions and videos together (so the identity of interlocutors and whether the interactions were directed or overheard could be checked) by the first and last authors.

## Interview data

126 adults (female = 83; mean age = 35.81, SD = 14.87), including all but 5 of the mothers of the recorded infants in both cohorts, were interviewed in 2019. During the interviews, adults were asked questions regarding their social, economic, and linguistic profiles, as well as those

of their family members, including all their infants. Interviewees could choose whether the questionnaire was administered to them in Spanish or Yucatec Maya. In the former case, the first author conducted the interviews; in the latter case, a local research assistant asked the questions from a previously verified translation. Both the first author and the research assistant were present during all interviews. See S1 Text for a copy of the questionnaire used and the coding of the responses.

All socioeconomic and demographic variables concerning the recorded infants and their families were obtained from these interviews. To assess the potential causal pathways driving any changes in linguistic socialization at the time of recording, for these analyses, we asked caregivers to answer in retrospective about their family situation (number of children, wage labour status, etc.) at the time of recording. Since some of the mothers' answers concerned the wage labor status of their husbands, we verified them by independently asking their husbands the exact same set of questions. When possible, their answers were further validated with basic demographic records collected at the time of recording.

## Informed consent

Informed, written consent was obtained from all adult participants included in the study and from all parents of the recorded children.

## Statistical approach

Bayesian inference was used for all statistical analyses. In a Bayesian framework, each model conditions its data on prior probability distributions and uses Monte Carlo sampling methods to generate posterior distributions for its parameters. The priors are the initial probabilities for each possible value of each parameter. Regularizing priors were adopted, which are more conservative than the implied flat priors of non-Bayesian procedures, in order to prevent the model from overfitting the data given the limited sample size [44]. Having fit alternative parameterizations for all models, we believe that the results presented below are qualitatively robust to changes in priors.

Even if the four villages were very similar, they differed in the extent to which they were represented. Random intercepts for "village" were included in all models to account for the nested structure of the data and associated clustering [44]. Before conducting our analyses, we checked for multicollinearity among predictors using the generalized variance inflation factor (GVIF). All GVIF values fell below the lowest commonly recommended threshold of 4, indicating that our models should not suffer from multicollinearity [45]. Our use of regularizing priors should also reduce variance inflation [46].

Parameter estimation was achieved with RStan [47], running three Hamiltonian Monte Carlo chains in parallel until convergence was suggested by a high effective number of samples and R^ estimates of 1.00 [44]. This entailed in some cases 5000 samples per chain and in others 10000. In the former case we used 1000 as warm-up and in the latter 2000. We also visually inspected trace plots of the chains to ensure that they converged to the same target distributions and compared the posterior predictions to the raw data to ensure that the models corresponded to descriptive summaries of the samples.

For model comparisons, we used Widely Applicable Information Criteria (WAIC) which provides an approximation of the out-of-sample deviance that converges to the leave-one-out cross-validation approximation in a large sample [48]. Analyses were performed in R 3.5.2 using the brms package [49,50]. We present a complete description and justification of the priors, model specifications, model comparisons and model coefficients in S1 Text.

### Ethical approval

All procedures involved in this study were reviewed and approved by the Ethics Commission of the University of Cambridge as well as the Ethics Committee of the University of Chicago. They are also in accordance with the 1964 Helsinki declaration and its later amendments or comparable ethical standards.

## Supporting information

**S1 Fig. Spanish level of the primary caregivers of the 14 infants recorded in Cohort 1 (left) and the 15 infants recorded in Cohort 2 (right).**
(DOCX)

**S2 Fig. Marginal posterior distribution of changes in the proportion of utterances in Spanish directed to infants (left), overheard by infants (middle) and received by infants overall (right) taking into consideration variation across the different villages.** The solid lines are posterior means and the shaded regions are 80% HPDIs.
(DOCX)

**S3 Fig. Posterior predictive distribution of the mean difference in number of utterances of each type received by the average child from 2007 to 2013, as obtained from the Zero-Inflated Poisson model including "Cohort" as predictor variable and the number of utterances of each type as response variable but excluding those infants from cohort 1 with ages of 23 or 24 months.** These were obtained by averaging from 12000 samples from the posterior distribution (setting the standard deviations for the varying intercepts to 0). From top-left to bottom-right: Directed input from primary caregiver, directed input from adults, directed input from children.
(DOCX)

**S1 Table. Descriptive statistics of the primary caregivers of the 14 infants recorded in Cohort 1 (left) and the 15 infants recorded in Cohort 2 (right).**
(DOCX)

**S2 Table. Descriptive statistics of infants recorded in Cohort 1 (left) and Cohort 2 (right).**
(DOCX)

**S3 Table. Descriptive statistics of all female adults interviewed in 2019 (n = 83).**
(DOCX)

**S4 Table. Descriptive statistics of all male adults interviewed in 2019 (n = 46).**
(DOCX)

**S5 Table. Nature of wage work of the adult males (n = 28) that participated in wage labour.**
(DOCX)

**S6 Table. Comparison of Zero-Inflated Poisson models predicting whether the number of utterances heard by infants in one hour had changed across Cohorts with equivalent Poisson models.**
(DOCX)

**S7 Table. Posterior predictive mean number of utterances of each type received by the average child from the Zero-Inflated Poisson model including "Cohort" as predictor variable and the number of utterances of each type as response variable.** These were obtained by averaging from 12000 samples from the posterior distribution (setting the standard deviations for the varying intercepts to 0).
(DOCX)

**S8 Table. Posterior predictive distributions for the changes in proportion of input in Spanish across cohorts for a child from an average village (in this context, average refers to setting the estimates of the standard deviations for the varying intercepts to zero).** "ll" refers to "lower limit" and "ul" to upper limit.
(DOCX)

**S9 Table. Comparison of Poisson models predicting the number of directed utterances received by infants in one hour.**
(DOCX)

**S10 Table. Self-reported beliefs about Spanish acquisition by the 94 adults that were fluent Spanish speakers.** The fact that the total is greater than 66 is because some women gave more than one answer to the question.
(DOCX)

**S11 Table. Beliefs by primary caregivers concerning the ways in which infants learnt the different languages.** Counts are the number of females that gave a specific answer to the question of whether they thought infants could acquire competences in Yucatec Maya in the same way as those in Spanish.
(DOCX)

**S12 Table. Raw averages across cohorts of the number of utterances per input type, both overall (rows 1–6) and in Spanish (rows 7–12).**
(DOCX)

**S13 Table. Posterior predictive distributions from the Zero Inflated Poisson models assessing changes in the number of utterances received by infants from their primary caregiver across cohorts and as a function of the number of utterances produced for a child from an average village (in this context, average refers to setting the estimates of the standard deviations for the varying intercepts to zero).** "ll" refers to "lower limit" and "ul" to upper limit.
(DOCX)

**S1 Dataset. Anonymised data.**
(CSV)

**S1 Text.**
(DOCX)

**S1 References.**
(DOCX)

## Acknowledgments

The authors would like to thank Prof Karen Kramer and Prof Robert Foley for helpful comments on the manuscript. The Chay-Cano Family, in particular Cornelio Chay Cano and Maria Leydi Hau Caamal for their assistance in the data collection process. All those that agreed to participate in the study.

## Author Contributions

**Conceptualization:** Cecilia Padilla-Iglesias, Amanda L. Woodward, Susan Goldin-Meadow, Laura A. Shneidman.

**Data curation:** Cecilia Padilla-Iglesias.

**Formal analysis:** Cecilia Padilla-Iglesias.

**Funding acquisition:** Cecilia Padilla-Iglesias, Amanda L. Woodward, Laura A. Shneidman.

**Methodology:** Cecilia Padilla-Iglesias.

**Project administration:** Laura A. Shneidman.

**Supervision:** Amanda L. Woodward, Susan Goldin-Meadow, Laura A. Shneidman.

**Visualization:** Cecilia Padilla-Iglesias.

**Writing – original draft:** Cecilia Padilla-Iglesias, Laura A. Shneidman.

**Writing – review & editing:** Cecilia Padilla-Iglesias, Amanda L. Woodward, Susan Goldin-Meadow, Laura A. Shneidman.

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
