## [Decision Letter · Decision Letter 0]

21 Jan 2021

PONE-D-20-39416

Why local languages disappear: a study of changing language input following market integration in a Yucatec Mayan community

PLOS ONE

Dear Dr. Padilla-Iglesias,

Thank you for submitting your manuscript to PLOS ONE. After careful consideration, we feel that it has merit but does not fully meet PLOS ONE’s publication criteria as it currently stands. Therefore, we invite you to submit a revised version of the manuscript that addresses the points raised during the review process.

We look forward to receiving your revised manuscript.

Kind regards,

Marcela de Lourdes Peña Garay, Ph.D

Academic Editor

PLOS ONE

Journal Requirements:

Reviewers' comments:

Reviewer's Responses to Questions

**Comments to the Author**

1. Is the manuscript technically sound, and do the data support the conclusions?

Reviewer #1: Yes

Reviewer #2: Yes

Reviewer #3: Yes

Reviewer #4: Partly

2. Has the statistical analysis been performed appropriately and rigorously? 

Reviewer #1: I Don't Know

Reviewer #2: Yes

Reviewer #3: Yes

Reviewer #4: I Don't Know

3. Have the authors made all data underlying the findings in their manuscript fully available?

Reviewer #1: Yes

Reviewer #2: Yes

Reviewer #3: No

Reviewer #4: Yes

4. Is the manuscript presented in an intelligible fashion and written in standard English?

Reviewer #1: Yes

Reviewer #2: Yes

Reviewer #3: Yes

Reviewer #4: Yes

5. Review Comments to the Author

Reviewer #1: This article reports changes in infant-directed speech that occurred between 2007 and 2013 in four Yucatec Mayan villages with populations between 400 and 600 inhabitants. The villages are close enough to the major urban centers of Cancun and Playa del Carmen to make it practical for men to travel to these towns for employment. The Yucatan Peninsula has seen major development in the international and national tourist economy for many decades, which attracts more villagers to these urban centers. This study offers a unique portrait of the impact that these changes have made to the language environment of Yucatec Mayan infants and thus to the forces responsible for language shift in the Yucatan.

The paper reports important research that contributes unique data to the study of language shift. Nevertheless, there are several issues the authors should address before their paper is published.

The title of the study suggests a causal connection between market integration and language change, and the first sentence of the Discussion (p. 8) states “Our results show, for the first time, that the quantity, type, source, and language given to Yucatec Mayan infants changed as the community became more connected with urban centers.” There are two difficulties with this claim. First, the paper does not supply details about the market integration taking place in the region between 2007 and 2013. Were there specific developments that took place or do the changes in input reflect prior changes or other factors? Second, the paper does not report any effects that changes in the input might have had on the children’s language production. Did the children continue acquiring Maya?

The main finding of the study is that “the amount of directed speech children heard in either language, particularly from other children, decreased over the 6-year period” (p. 5). This result is at odds with explanations of language shift in that the Yucatec communities aren’t shifting languages so much as refraining from talking to their children. The authors do not offer an explanation for the reduction in speech to children. Does the reduced speech to children reflect the forces of market integration or the effect of other changes taking place in the Yucatan?

Restricting attention to the speech directed to the children, the authors report an increase in the proportion of Spanish utterances directed to the children (p. 5). The overheard speech did not exhibit a similar shift, and thus the result leads to the question of what factor would explain this shift? The authors phrase the issue (p. 6) as one of decreased use of Maya rather than an overall decrease in their speech to infants. This perspective is at odds with the statement (p. 5) that the parents also decreased their use of Spanish.

The ethnographic description of the households and villages is too thin to assess the impact of other factors and thus evaluate the significance of the four factors identified in the study. One omitted factor is the introduction of satellite tv to Mexican villages. Many remote Mexican villages now have access to cable tv, and households with cable tv watch it much of the day, but especially in the evenings. Television dissolves the social fabric by subtracting from the time that families previously spent gossiping in the local language with friends and family. The paper should state the number of homes that had tv in the two cohorts.

The introduction of cable tv offers a direct tie to the forces of market integration in that the men who work in the tourist sector would see television as a marker of social status and want to take television back to their own village. Men with cable tvs would have more status than men who just worked in the milpa and so television would take over the village. This hypothesis recognizes a pathway for importing urban behavior to the home village leading to a shift in culture that underpins the shift in language. A factor like “the general devaluing of the native language” does not identify the source of such a devaluing and how it becomes general. The paper does not identify a pathway that ties language devaluation to market integration.

The paper mentions in passing a more sinister factor in the absence of Yucatec teachers and language materials in the village schools (p. 10; Supplementary text, p. 2). The possibility has to be admitted that the absence of such support is a deliberate policy on the part of the national government to suppress the use of local languages. Many indigenous teachers are sent to teach in urban areas, while teachers from urban areas are sent to teach in rural schools. Teachers from outside the village are more likely to model intolerant attitudes towards the local “dialects”. It would be helpful if the paper supplied some information about where the village teachers come from and their attitudes towards Yucatec.

The authors should clarify the following points:

1. Participants: The paper should state the number of participants, if any, who were included in both cohorts. Including the same families in both studies might lead to an observer effect and possibly depress the number of utterances they produced. How many of the original families participated in the interviews? The utterances were classified by speaker: “primary caregiver (mother), other adult (over 11 years), or other children” (p. 11), but the paper only provides information about the primary caregivers (S1 Table) and the target infants (p. 11; S2 Table). It should provide similar tables for the other speakers. All of the participant information should be presented in one place. In the discussion on p. 9 the paper states it is possible “that older children are focusing more on schoolwork and less on sibling care.” This should be obvious from the number of older children present in the two cohorts.

2. Language samples: The families were “recorded for 60 minutes in their homes in natural interaction” (p. 11). The authors should add more information about these recording sessions such as whether they were made inside or outside the house, and the proportion of time spent looking at books. Changes in the use of books between the two cohorts might account for the changes in the input.

3. Maya versus Spanish: “Utterances were classified as Spanish only if the entire utterance was in Spanish. Utterances where code-switching between languages occurred were not included in this category and neither were utterances in which Spanish words were used as nicknames” (p. 11). It is unclear whether utterances with code-switching were excluded entirely or coded as Yucatec. The discussion of Figure 3 (p. 8) defines the proportion of Spanish utterances as the “Number of directed utterances in Spanish out of the total number of directed utterances” so it appears that the authors coded the utterances as either entirely Spanish or other. This interpretation is at odds with the statement “while infants in cohort 1 heard an average of 397.3 Mayan utterances per hour in directed speech, infants in cohort 2 children heard only an average of 92.51 Mayan utterances” (p. 6). On p. 5 the paper states “To examine whether the language of input shifted from Yucatec Mayan to Spanish over the 6-year period, we fitted Bayesian logistic mixed models to the proportion of input in Spanish (vs. Mayan) that infants received in child directed and overheard speech.” The paper should clarify what was counted as a Mayan utterance, and address the question of whether the code-switched utterances were excluded from both the Mayan and Spanish counts.

Without knowing how the proportion of code-switched utterances changed between the two cohorts, it’s impossible to tell if the number of Spanish utterances changed significantly. The paper should report the number of code-switched utterances. The code-switched utterances should have been analyzed for the number of utterances with verbs in Spanish or Yucatec. In fact, the proportion of utterances containing Yucatec or Spanish verbs would be a more sensitive measure of language shift than the proportion of Spanish utterances because it would obviate the need to decide whether or not individual words were Spanish. An increase in the number of utterances consisting entirely of Spanish nouns and noun phrases (un carro, chicle, galleta) would not suggest a significant shift in language.

4. Villages: The paper states that the recordings were made in four villages (p. 10), and that the villages were very similar (p. 12). The Supplementary text adds information about the populations (p. 2). All information about the villages should be presented in one place in the paper. The paper should provide more details about the relative levels of roads and schools in each village, the proportion of the men who participated in wage labor from each village, and the number of participants from each village in each cohort. These details would help readers understand the degree to which the changes took place regionally rather than principally in one or two villages.

This is a solid investigation and I encourage the authors to add further details to their paper.

Reviewer #2: The paper examines factors involved in language shift by analyzing interview data, family histories, and transcripts of natural data (to assess input) over a 6-year period in 4 Yucatec Mayan villages where language shift to Spanish is in process. The analysis supports the central argument, that Maya language input has decreased due to social changes that are not explicitly related to language but rather to other social changes (e.g. increased numbers of Mayans working for wages in Spanish-speaking cities, decrease in the availability of older siblings to care for young children because they now attend formal schooling at higher grades than previously) which directly affect language input. The paper is clearly researched and argued; the data are clearly presented and all analyses explained thoroughly and well-documented. The supplementary materials provide robust data supporting changes in education, in the caretakers, and in the language used by them with the children, and in the fathers' earning sources. (I did not see data supporting an increase in market goods or directly linking an increase in wage labor to Spanish, maybe I missed it.)

The finding that Mayan communities find Spanish more practical than Yucatec Maya is not surprising or novel, given similar findings in most shifting ecologies; the novelty is perhaps a bit overstated in the paper. The significant contribution of this paper is the very rigorous way it has determined the attitudes and beliefs of the speaker communities and the actual input the children, in both directed and overheard speech. It is also provides a model for future work in other communities to understand the causes of shift.

Minor revisions:

The relevant social and demographic changes discussed here are encapsulated by the authors with the term market integration, and I would recommend that the authors define what they mean by market integration in this paper early on, as it is used in a non-standard way, OR change the terminology to specify increased contact with urban centers (which is specified on p. 8) OR increased education/wage labor/market goods (p. 4). The discussion of the relevant urban contact is provided under Materials & Methods on p. 16 of the manuscript. A sentence definining what is meant in the beginning of the paper would be welcome.

I would further note that it is odd to think about periods of market integration:

"during periods of market integration, both parents and older siblings of young children in small-scale communities are more likely than previous generations to have had access to formal schooling provided by the majority community" (p. 2).

Market integration is not usually conceptualised as a transitory state. Rather, markets usually start out not integrated and then improvements in road conditions or whatever barriers there were to integration would make them better integrated. So here my sense is that the intent of the authors is a before/after comparison, and the wording could be tweaked.

Reviewer #3: see attachment--uploaded attachment would exceed character count, I think;

The paper presents one very significant finding, based on a comparison of two different cohorts of Maya Yucatec speakers in two different periods of language-acquisition studies separated by 6 years. I found this main empirical finding striking and important (although not without some questions). I think the work should be published if only to disseminate this well founded and potentially useful result. It is, in brief, that over the very short 6 year span, there was significant change in the sort of direct language input infant Yucatec Maya language learners received. It went from being largely in Maya to almost entirely in Spanish (with some empirical caveats I will shortly mention).

Reviewer #4: [see attachment] [see attachment][see attachment][see attachment][see attachment][see attachment][see attachment][see attachment][see attachment][see attachment][see attachment][see attachment] - Sorry it wouldn't let me submit without 200 characters here!

6. PLOS authors have the option to publish the peer review history of their article (what does this mean?). If published, this will include your full peer review and any attached files.

Reviewer #1: **Yes: **Clifton Pye

Reviewer #2: No

Reviewer #3: **Yes: **John B. Haviland

Reviewer #4: No

---

## [Author Response · Author response to Decision Letter 0]

24 Feb 2021

Please see the attached "Response to reviewers.docx" document.

---

## [Decision Letter · Decision Letter 1]

6 Apr 2021

PONE-D-20-39416R1

Changing language input following market integration in a Yucatec Mayan community

PLOS ONE

Dear Dr. Padilla-Iglesias,

Thank you for submitting your manuscript to PLOS ONE. After careful consideration, we feel that it has merit but does not fully meet PLOS ONE’s publication criteria as it currently stands. Therefore, we invite you to submit a revised version of the manuscript that addresses the points raised during the review process.

We look forward to receiving your revised manuscript.

Kind regards,

Marcela de Lourdes Peña Garay, Ph.D

Academic Editor

PLOS ONE

Additional Editor Comments (if provided):

All reviewers and I find the subject presented in this manuscript very interesting and original. However, there are still important issues that must be clarified to advance with the publication consideration in PLOS ONE.

I agree with reviewer 4 that providing more data supporting the statement that socioeconomic change in the analyzed periods, disentangling the role of market integration and secondary schooling in the linguistics changes, and clarifying the concept of language extinction based on current models, would confer the manuscript greater impact.

In addition to te comments of reviewer 4, the suggestion of reviewer 3 about to tone down or rephrase the claims made on the basis of the survey data, may benefit the clarity of the ms.

Reviewers' comments:

Reviewer's Responses to Questions

**Comments to the Author**

1. If the authors have adequately addressed your comments raised in a previous round of review and you feel that this manuscript is now acceptable for publication, you may indicate that here to bypass the “Comments to the Author” section, enter your conflict of interest statement in the “Confidential to Editor” section, and submit your "Accept" recommendation.

Reviewer #1: (No Response)

Reviewer #2: All comments have been addressed

Reviewer #3: All comments have been addressed

Reviewer #4: (No Response)

2. Is the manuscript technically sound, and do the data support the conclusions?

Reviewer #1: Yes

Reviewer #2: Yes

Reviewer #3: Partly

Reviewer #4: Partly

3. Has the statistical analysis been performed appropriately and rigorously? 

Reviewer #1: I Don't Know

Reviewer #2: Yes

Reviewer #3: I Don't Know

Reviewer #4: I Don't Know

4. Have the authors made all data underlying the findings in their manuscript fully available?

Reviewer #1: Yes

Reviewer #2: Yes

Reviewer #3: No

Reviewer #4: Yes

5. Is the manuscript presented in an intelligible fashion and written in standard English?

Reviewer #1: Yes

Reviewer #2: Yes

Reviewer #3: Yes

Reviewer #4: Yes

6. Review Comments to the Author

Reviewer #1: Manuscript #: PONE-D-20-39416.R1

Title: Changing language input following market Integration in a Yucatec Maya Community

I thank the authors for their attentive responses to my comments. They have responded well to my queries and I would now accept their paper for publication.

The study reports two key findings, but the interpretation of the results remains problematic due to the inherent complexity of sociolinguistic data. I am especially concerned about the explanation for the decrease in the number of utterances between cohorts (S7), which the authors link to changes in the villages’ market integration. The alternative is that this difference is due to some other difference between the cohorts. The authors’ responded to my initial concern about this decline by noting that:

"In fact, changes in directed input were driven (fully) by decreases in input in Yucatec Maya. Input in Spanish was very rare in the first cohort (mean of 7 utterances per hour) and increased in cohort 2 (80 utterances per hour). Thus, two changes were taking place simultaneously: a reduction in overall speech directed to children and a shift in the language that was used (from Maya to Spanish) in instances where speech was directed to children."

The description in the text should be modified to clarify the fact that the decline in input reflected decreases in Maya utterances. The authors might consider replacing table S7 with a table that just reports the number of Maya utterances in directed input across the two cohorts. I combined the numbers in S7 and S8 into a single table in order to simplify the comparison:

Number of utterances S7 Proportion of Spanish S8

mothers Other adults children mothers Other adults children

Cohort 1 99.55 36.86 346.69 10% 1% 27%

Cohort 2 76.81 37.79 165.72 29% 66% 61%

I don’t see how to reconcile the numbers in the authors’ response with the numbers in S7 and 8. For Cohort 1, 10% of 99.55 ~ = 9.9 utterances (close to 7 utts/hr.) For Cohort 2, 29% of 76.81 ~ = 25 utterances (not close to 80 utt/hr.) This confusion suggests it would help if the authors supply the number of utterances in Maya and Spanish as well as the proportions. It’s good practice to provide both numbers and proportions. My calculations suggest that the mothers in Cohort 1 produced a mean of ~90 Maya utterances, and mothers in Cohort 2 produced a mean of ~51 Maya utterances, while their Spanish utterances increased from 9.9 to 25.

These numbers still don’t account for the decrease in Maya utterances between cohorts assuming that the children’s basic interactions wouldn’t change between cohorts. This is one of the findings that as the authors state “Our results show, for the first time, that the quantity, type, source, and language input heard by Yucatec Maya infants changed as the communities became more connected with urban centers, and hence education, wage labour and market goods became more accessible to their inhabitants.” The authors link the decrease in speech from other children to a decrease in time the older children spent with the target children due to schooling (p. 9 and response to reviewer). This explanation doesn’t explain the decrease in the mother’s directed speech between cohorts.

If the target children’s behavior changed between cohorts then the target children’s behavior, not the behavior of other speakers, would account for changes in the number of utterances. One change in the target children’s behavior might reflect the difference in the children’s ages between the cohorts (Median 22 mos in Cohort 1 and 18 mos in Cohort 2, S2). These differences could correspond to differences in the mean lengths of the target children’s utterances from a one-word stage to a two-word stage. Thus, differences in the mothers’ CDS between the cohorts might reflect differences in the target children’s language production rather than changes in market integration. Or both factors might be at work.

In their response to my previous question about the children’s language production, the authors state that they “did not quantitatively assess children’s proficiency on each of the languages…” This observation makes sense from the perspective that CDS is the primary driver of the children’s language development, but ignores the point I raise here that the children’s language production might account for the decrease in the mothers’ directed speech in Maya. It would be a simple matter to calculate the target children’s mean length of Maya and Spanish utterances in words across the two cohorts to test this hypothesis. An alternative would be to acknowledge the target children’s language as a possible explanation for the decrease in speech between cohorts. It would help to add the authors’ new observations about the children’s current use of Maya and Spanish in order to add a valuable perspective on the limited effects of market integration. It might yet prove to be the case that despite changes in the economy, the villages will persist in their use of Maya.

Reviewer #2: not applicable. I have no comments to make to the author and am annoyed that I need to fill this box in, having already indicated above that the changes addressed my concerns.

Reviewer #3: One of my main suggestions, namely to tone down or background the claims made on the basis of the survey data, the authors chose to ignore, and it is thus difficult for me to reassess the paper given my reservations about its original design, which remains largely unchanged.

Reviewer #4: (No Response)

7. PLOS authors have the option to publish the peer review history of their article (what does this mean?). If published, this will include your full peer review and any attached files.

Reviewer #1: **Yes: **Clifton Pye

Reviewer #2: No

Reviewer #3: No

Reviewer #4: No

---

## [Editor Report · Decision Letter 2]

26 May 2021

Changing language input following market integration in a Yucatec Mayan community

PONE-D-20-39416R2

Dear Dr. Padilla-Iglesias,

We’re pleased to inform you that your manuscript has been judged scientifically suitable for publication and will be formally accepted for publication once it meets all outstanding technical requirements.

Kind regards,

Marcela de Lourdes Peña Garay, Ph.D

Academic Editor

PLOS ONE

Additional Editor Comments (optional):

The authors rewrite the manuscript to answer all the concens made by the reviewers.

In the current state the manuscript provides new data and may inspire further discussion and research.
---

## [Editor Report · Acceptance letter]

2 Jun 2021

PONE-D-20-39416R2 

Changing language input following market integration in a Yucatec Mayan community 

Dear Dr. Padilla-Iglesias:

I'm pleased to inform you that your manuscript has been deemed suitable for publication in PLOS ONE. Congratulations! Your manuscript is now with our production department. 

Kind regards, 

on behalf of

Dr. Marcela de Lourdes Peña Garay 

Academic Editor

PLOS ONE